# The Dorsomedian Ligamentous Strand: An Evaluation In Vivo with Epiduroscopy

**DOI:** 10.3390/medsci10010018

**Published:** 2022-03-07

**Authors:** Maurizio Marchesini, Eleonora Schiappa, William Raffaeli

**Affiliations:** 1Mininvasive Surgery Department, Unit of Pain Medicine IRCCS Maugeri Pavia, 27100 Pavia, Italy; 2I Service Anesthesia, Critical Care, Azienda Ospedaliero Universitaria Parma, 43121 Parma, Italy; schiappa.eleonora@gmail.com; 3ISAL Foundation, Institute for Pain Research, 47900 Rimini, Italy; wraffaeli@yahoo.it

**Keywords:** interventional pain procedure, dorsomedial ligamentous strand, epidural space, epiduroscopy

## Abstract

Several anatomical studies have described the morphology of the spinal space; however, researchers do not all agree on the presence of the dorsomedian ligamentous strand (DLS), which divides the epidural space. The possible existence of this structure still influences some clinical practice, such as locoregional anesthesia and pain therapy. Since the number of procedures occurring inside the epidural space have increased, this study’s primary objective was to describe the composition of this space through epiduroscopy. We conducted a retrospective analysis of video recorded during epiduroscopy. Two independent doctors performed blind analyses of morphological aspects of peridural space visualized during the procedure in each patient for the maximum possible extension depending on the underlying pathology in the tract from S1 to L1. We enrolled 106 patients who underwent epiduroscopy; 100% of patients presented no medial longitudinal segmentation dividing the epidural channel at any level of the spinal tract investigated, including in the epidural space with pathological fibrotic scars and in those with no adherence. The main finding of our study was the visual absence of any anatomical structure dividing the epidural channel. We report that in vivo, in our experience, with direct epiduroscopy, the DLS is not visible.

## 1. Introduction

Since the 1900s, many anatomical studies have attempted to describe the structural and morphological features of the spinal space [1]. Despite the many procedures developed over the years using the anatomical and functional specificity of this space and surgical practices on its components, little is known about the composition of the epidural space. Although this space is a common site of interventional and surgical procedures, few studies have investigated this anatomical compartment.

In addition to orthopedic surgeons, neurosurgeons, and anesthesiologists, pain therapists, radiologists, and neurologists work on the epidural space for diagnostic and therapeutic purposes by applying anatomical landmarks based on dated studies [1].

A division of the epidural space into two compartments has been described since 1970 using names such as “plica mediana dorsalis” [2] or “dorsomedian dural fold” [3]. This segmentation is considered real despite some papers that encourage reflection on the evidence of this “dorsomedian ligament” [4]. Analyzing the studies conducted on this anatomical structure, which specify its morphological characteristics, indicates that, while some support its presence based on anatomical verification or CT documentation, others deny it, considering it an autoptic artifact.

The existence of a median segmentation has been controversial since its first description in 1963 [5]. Luyendijk first described the so-called “plica mediana dorsalis” using epidurography and then confirmed its discovery, in 1976, by photographs taken during laminectomy [2]. In 1980, Husemeyer and White published a study showing this segmentation (defined as a dorsomedian dural fold), having injected polyester resin into the epidural space of adult cadavers [3]. Through endoscopy, Blomberg [6] et al. described a band of connective tissue on the dorsal midline of the epidural space in 48 analyzed cadavers. Furthermore, in 1989, Savolaine [7], using CT to examine the epidural space, identified, in all 40 patients studied, that the posterior epidural space was divided by a DLS and supplementary radial connective tissue. However, a debate was immediately sparked concerning whether the DLS existed or was only an artifact. In 1991, Hogan [8], through analyzing cryomicrotome sections of the lumbar spines of 38 frozen cadavers (obtained within 15 h of death), showed that no median ligamentous strand appeared in any of them. The ligament would probably reflect an artifact due to the dural tension (in the presence of an unnatural transmural pressure during distention of the epidural space).

Introducing endoscopic procedures did not improve this uncertainty. Blomberg documented the presence of connective tissue between the rear of the dural sac and the yellow ligament defined as “dorsomedian ligamentous strands”, which, due to the traction exerted on the dura, produced the plica mediana dorsalis [6]. This anatomical theory was countered by cadaveric anatomical studies [9] and using endoscopic procedures that improved the resolution power with adequate optical sources over the years. This debate is not only theoretical since it reflects on the choice of procedure: anesthesiological selective block [10], neuromodulation/neurostimulation methods for pain therapy [4], and minimally invasive spinal surgery [11]. Since epidural space procedures to relieve pain and provide anesthesia are gaining broader therapeutic value, and the use of therapeutic epiduroscopy is increasing [12,13,14], we believe it is useful to review the composition of the spinal space.

This study aims to highlight the presence or absence of signs supporting the existence of the ligament by in vivo epiduroscopic analysis.

## 2. Materials and Methods

The epiduroscopy procedures were performed in the context of the PainOmics study. The authorizations of the ethics committee and informed consent signed by all the patients allowed for the analysis of video recorded for retrospective study and clinical reasons. We made a retrospective review of the video recorded during the execution of epiduroscopy procedures performed from 2014 to 2016 in a single center by a single operator.

Patients were enrolled with the following inclusion criteria:-Age 18–85 years-Patients with failed back surgery syndrome (FBSS) and spinal stenosis-Able to sign an informed consent form

The following exclusion criteria were applied:-Patients with clinically unstable disease (all morbid forms whose treatment is not stable over time but requires continuous pharmacological and dosage adjustments or require further investigations)-Patients who have been diagnosed with severe neuropsychiatric disease and have received pharmacological treatment under constant medical supervision for which there is a contraindication for invasive treatments-History of vertebral fractures-Tumors or infections affecting the spine-Visual impairment (glaucoma, diabetic retinopathy)-Chronic primary or secondary headache-Pregnancy-Coagulopathies (INR > 1.5)

General medical aspects (unstable primary pathologies recommend a waiting attitude for non-urgent procedures) and specific clinical aspects for the epiduroscopy maneuver defined the exclusion criteria to limit the risks of infectious and neurological complications as reported in the literature [15].

For each patient, we analyzed images related to the macroscopic morphology of the contents of the epidural space from S1 to T12–L1. Particularly, we searched for a DLS (and whether it appears as a small embryo), which should divide the epidural space according to previous literature [13].

The channel was explored within the most extensive tract from S1 to L1 allowed by the patient’s pathology.

For each video, analysis was performed independently by two clinicians who were experts in epiduroscopy (more than 100 procedures each). In case of non-agreement, the video was reviewed by the most expert operator (Dr. Raffaeli) for a third opinion.

For a detailed description of the epiduroscopy, refer to previously published articles [15].

## 3. Results

We analyzed videos of 106 patients (41 men and 65 women) with a median age of 58 years.

The diagnoses that led to the indication of periduroscopy are summarized in Table 1.

We included 11 cases (10%) in the data analysis where it was impossible to explore at least up to L2.

Cases where it was impossible to explore at least up to L3 (4%) were excluded from data analysis.

The macroscopic composition of the epidural space includes the epidural channel, dura, roots, and yellow ligament.

Independently from the pain syndrome described in the diagnosis, 100% of patients presented no medial longitudinal structure dividing the channel.

Due to an abnormal laxity of the dura in one case (0.16%), we observed a segmentation-like feature secondary to the traction exerted by the video guide and Fogarty balloon working, which induced a shake.

The video analysis enabled some original images to be obtained that could explain how it could be possible to find morphological abnormalities responsible for false radiological findings in some cases.

Figure 1 enables visualization of the composition of the normal dura and the other structure and directly indicates the absence of any median strand.

Figure 2 shows the lifting of the meningeal dura for dural mobility abnormalities with an underlying lifting of the spinal root, which determines a pseudo-lifting of the dura, potentially resulting in a false diagnostic radiology interpretation of a median ligament. Moreover, the presence of a Fogarty catheter inside the bed of the dura in the midline causes lifting of the dura mater in relation to the root below, which can mimic a false longitudinal ligament in radiodiagnostic imaging (Figure 3). Morphological analysis of the pathological spaces also showed that, in these cases, no segmentation occurred. The left side of Figure 4 shows a fibrotic septum grafted on the dura. However, underlying dural bases exhibit no segmentation.

## 4. Discussion

Our previous studies have highlighted that epidural space has different morphological characteristics based on the presence or absence of alterations of the dural cloth, roof, vessels, and channel. This study confirms, in a significant number of patients, what we had previously reported [14]: no connective septum exists in the epidural space. None of the patients in our study showed a medial longitudinal ligament.

Our in vivo experience does not confirm the findings arising from the initial endoscopic procedures by Blomberg [6] and suggests that the DLS may be a cadaver artifact.

In the first diagnostic part of the procedure, a low volume of saline (between 20 and 100 mL) was used; we suppose that this small volume cannot be hidden or modify the presence of a solid structure from yellow ligament to dural meningeal pannus.

Thus, an even larger volume (300–500 mL), as used during the therapeutic phase of adherence lysis, probably cannot change the anatomy of a hypothetical DLS.

The presence of fibrotic septa, without apparent cause and limited to a single vertebral body, was detected in a few cases. In our video review, we found one patient with a segmentation-like feature. This appeared to be the result of a raising of the dura due to loss of its elasticity and the presence of a hyperpressure on the dural cloth (Figure 2), as previously reported [16].

We consider that the descriptions given by many authors of images obtained through endoscopy or radiological imaging can be attributed to errors due to the technology of cadaver fixation, according to Morisot [17]. Alternatively, they may be interpretations of a dorsal membrane in pathologies (i.e., compartmental fibrosis in stenosis) that present features that CT cannot detect. A polyester resin print of epidural space with injection in cadavers has shown only in a few cases a faint dorso-median fold of the dura [18].

The study was carried out more accurately through the injection of contrast medium after insertion of epidural catheter. The execution of both CT and spiral CT and MRI showed, but only at the thoracic level, the presence of a fold, however, with an extensive presentation variable [19].

At the lumbar level, there are not, to our knowledge, similar rigorous studies. The most recent literature only shows anectodical descriptions of dorsal plica during diagnostic radiological studies (such as for oncological pathologies) [19].

In accordance with Hogan [9], we believe that the membrane could be attributed to raising of the dural cloth due to pressure secondary to the use of rigid instruments on the dura (Figure 3). Moreover, in the presence of tractions secondary to pathological events (such as post-surgical or post-inflammatory fibrosis) on the dura mater, the membrane could be raised by generating, with its “sail” shape, a segmentation-like feature [16] (Figure 4). Hogan claimed that in addition to the absence of any “fibrous barrier across the intervertebral neural canals” a narrow fibrous band could sometimes be seen adjacetn to the superior edge of pedicle epidural space … and epidural space is widely open on its lateral aspect” [8]. Asato published a paper indicating that no obstacle existed to the spread of epidural solution due to a median epidural septum, and the epidurography, performed in seven patients with unilateral epidural block, showed that the cause of the unilateral block was the placement of the catheter into the anterior or transforaminal epidural space [18]. Transversal segmentation, which divides the space into two sections, can be present in subjects with diseases secondary to chronic polyfactorial stenosis or where previous spinal surgery justifies abnormalities in fluid distribution, based on the CT findings by Savolaine [7].

## 5. Conclusions

Our video analysis involved over 100 patients, a significant number for a “subjective” visual analysis such as the one we performed. This evaluation cannot be definitive but raises further doubts concerning the existence of the DLS in vivo, particularly due to the total absence in any patient of signs of its presence.

Further studies in vivo, with high-resolution neuroimaging (for example, 3 Tesla MRI) or with fresh cadavers, could support or reject the hypothesis of the existence of a structure partitioning the posterior epidural space.

## Figures and Tables

**Figure 1 medsci-10-00018-f001:**
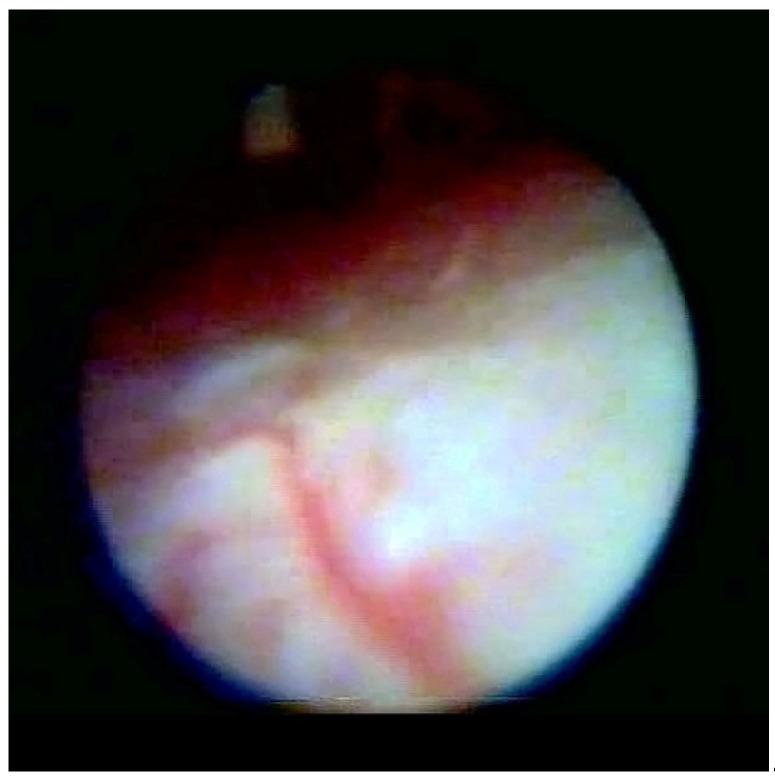
Normal dura in epiduroscopy.

**Figure 2 medsci-10-00018-f002:**
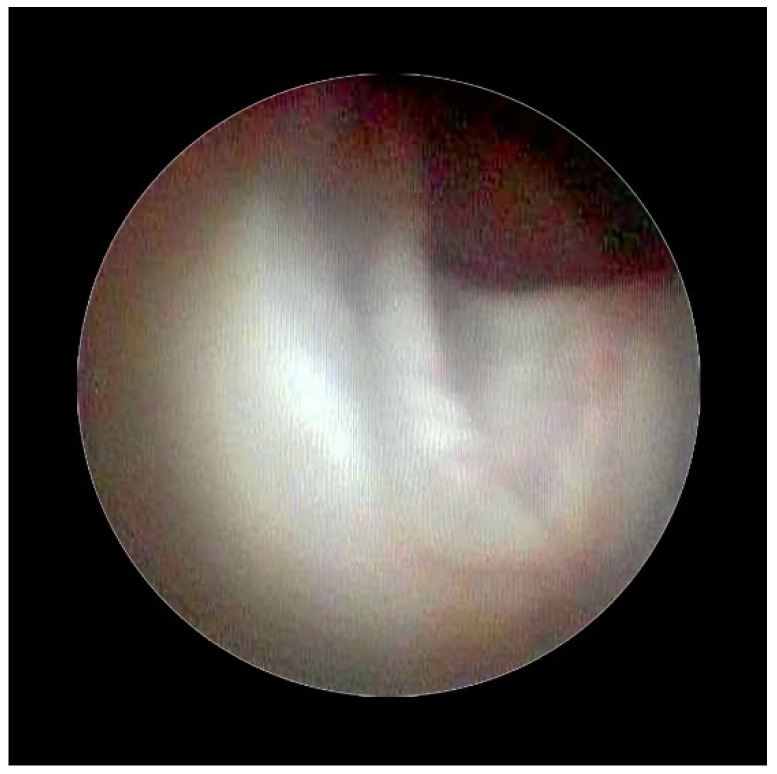
Normal dura, root lifting.

**Figure 3 medsci-10-00018-f003:**
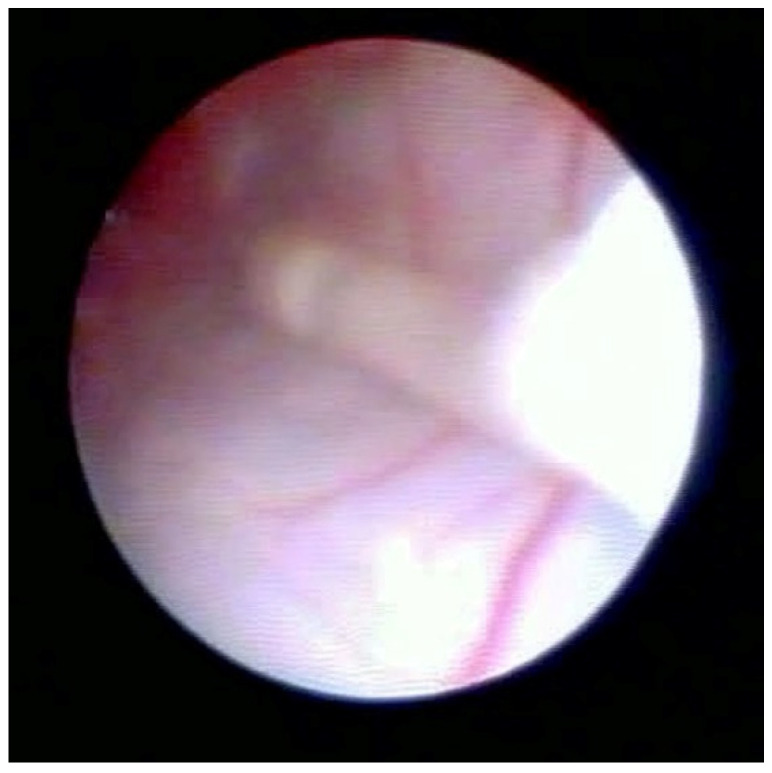
Fogarty catheter close to dura.

**Figure 4 medsci-10-00018-f004:**
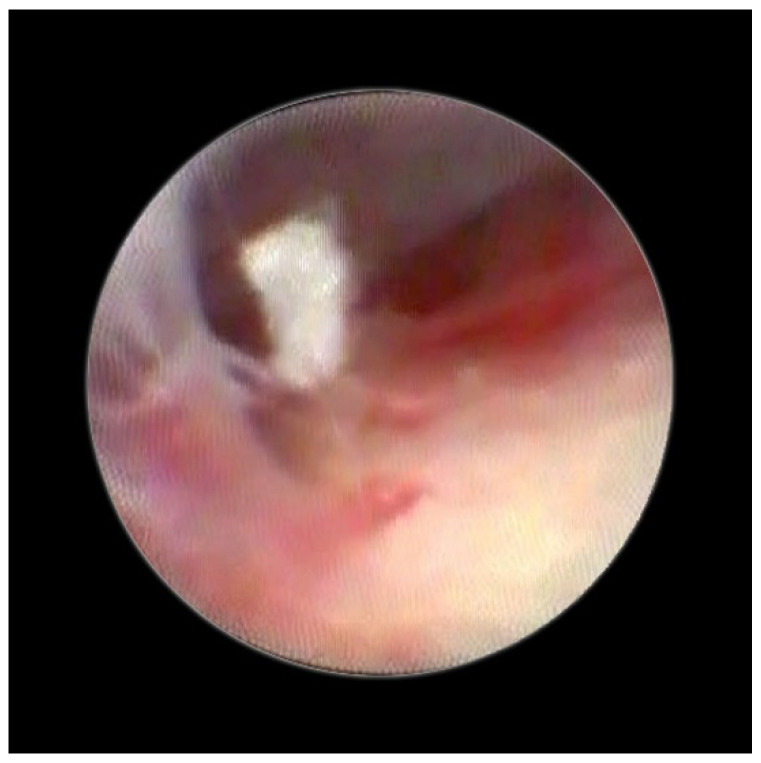
Fibrotic septum in epidural space.

**Table 1 medsci-10-00018-t001:** Description of types of chronic persistent lumbar pain with or without radicular pain.

Diagnosis	Number *N* (%)
Peristent back pain secondary to spinal stenosis (ostogenic, discopaty)	40 (37.7%)
Persistent pain secondary to spine surgery (failed back surgery syndrome)	60 (56.7%)
Persistent back pain with no instrumental diagnosis	4 (3.8%)
Pain from extra-spinal conditions (traumatic sacrococcygeal pain) without documented lumbar disease	2 (1.8%)

## Data Availability

The data presented in this study are not available on request from the corresponding author.

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
