# Peer review of "The Dorsomedian Ligamentous Strand: An Evaluation In Vivo with Epiduroscopy"

_medsci, 2022, doi:10.3390/medsci10010018_

Round 1
Reviewer 1 Report
I have no new comments.
Author Response
Many thanks for your approval.
I tried to make it easier to read by further correcting the syntax and adding a part of the discussion with some additional references, also thanks to your contribution.
Best regards.
Maurizio
Reviewer 2 Report
- In page 2 line 81, the authors are encouraged to show the full version of the abbreviation FBSS
- Can the authors briefly explain the reason for major exclusions mentioned in lines 84-89?
- The manuscript needs to be more cohesive in terms of writing, but other than that it is a great rebuttal study for absence of DLS in vivo.
Author Response
Many thanks for your efforts to make this paper better.
I proceeded to add the full version of FBSS.
I briefly justified the reasons for the exclusion criteria by referring back to the complication literature.
I tried to make the article more readable by slightly broadening the discussion and bringing more bibliographical references.
Kind regards
Maurizio
Reviewer 3 Report
The procedures were performed and s, the videos were analyzed by experts and when there was no consensus by a third party. This description of the absence of the plica mediana dorsalis is very interesting, since to my knowledge it is the first time that it has been intentionally searched for in vivo. There could be doubts about the possibility that during epiduroscopy the ligament was unintersected, but this is unlikely since it was not found in 100% of the cases.
The article is well written and clinically useful. It seems to me that the discussion could be more interesting, especially a discussion of magnetic resonance imaging (image of the plica is sometimes found especially in tumor lesions).
Author Response
Dear Reviewer,
many thanks for your efforts to make this paper better.
I have added some points in the discussion to make it more stimulating as you advised, both in confirmation and refutation of the hypothesis expressed. Help open again a debate that has been inactive for over 20 years, thanks to new technologies, is the paper's primary purpose).
Unfortunately, many works, as mentioned, are dated, however, I have reported some points, extending the bibliography as well, to the radiological descriptions (CT and MRI) of the dorsal ligament.
Unfortunately, as I report in the discussion, the studies that, as you rightly suggest, remark the presence of dorsal ligament in tumor lesions not as the primary aim of the research but as a collateral description.
In my opinion, this way to report the dorsal ligament is less systematic and less repeatable. It is difficult to define if in MRI vision the object described is the plica or a component of the tumor lesion or a reaction to it. This way to briefly mentioned as a general description (both radiological and surgical) lacks evidence as opposed to a further article cited that at the dorsal level draws the study precisely with this purpose.
best regards
Maurizio
This manuscript is a resubmission of an earlier submission. The following is a list of the peer review reports and author responses from that submission.
Round 1
Reviewer 1 Report
I thank the Editor in Chief for the opportunity to review the article entitled "The Dorsomedian Ligamentous strand: an evaluation in vivo with epiduroscopy". The article by Marchesini et al attempts to highlight a debated aspect in the anatomical field through an evaluation process based not on the analysis of the anatomy in the cadaver but through direct visualization through an interventional procedure such as periduroscopy. As correctly pointed out by the authors in the introduction, previous studies had already questioned the presence of the dorsomedian ligamentous strand. Still, its existence had never been sought by endoscopic maneuvers in vivo. This approach appears exciting and innovative, even if burdened by some limitations. The number of periduroscopic procedures appears to be congruous. The method applied seems to guarantee an evaluation by several operators to ensure fair repeatability, albeit always dependent on the operator. The main limitations of the study lie in two main aspects, as correctly cited by the authors: - The method is not created to highlight specific anatomical structures; the visualization is always dependent on the anatomical conditions of the patient and the operator's skills during the procedure. - The review of the videos, even if done by two expert operators, presents a significant subjectivity. The presence or absence of the dorsomedian ligament remains a topic of anatomical debate. As written by the authors, this paper does not intend to give a definitive answer. Still, it represents a new way of viewing the epidural space, of evaluating its anatomy not only in a therapeutic/diagnostic but also in an anatomical sense, proposing a new way of highlighting the problem. He adds a different point of view to the discussion, highlighting his absence in the conclusions. The proposed iconography appears adequate, and the linguistic structure has received a correct proof review. Minor concern- The article contains an excess number of self-citations (4/16). Since it is over 20 years old, the reference number 16 (i.e., Raffaeli W, Balestri M. Epiduroscopy: Preliminary reports-technical notes. The pain clinic 1999; 11: 209-212.), Could be removed.
- Some plurals must be corrected in the text. See below.
- Line 106. Correct the word 'image' as 'images'.
- Line 109. Correct the word 'structure' as 'structures'
- Line 117. Correct the sentence 'There were not any segmentation' (there were not any segmentations? there was not any segmentation?).
- Line 147. Please, correct 'compartmental fibrosis in stenosis' as 'compartmental fibrosis in a stenosis'.
- Line 166. Please, correct 'one hundred of patients' as 'one hundred patients'
- Finally, plagiarism check is ok (see attachment).
My best regards

Reviewer 2 Report
The authors report their negative results in 100 patients undergoing endoscopic exploration of the epidural space.
After performing over 2000 laminectomies myself, I concur with the authors that I have also not observed the described dorsomedian ligamentous strand.
Unfortunately, this study does not bring anything new to the literature.
As a side note, the English can be significantly improved.
Reviewer 3 Report
The presence of the dorsomedian ligament is evaluated in a cohort of 106 patients who underwent epiduroscopy. In none of the patients was this strucutre found. In the literature other studies using various methodologies have conflicting results on the existence of this structure. In the present study the authors suggest that the the dorsomedian ligament may be an artificial strucutre, made by exerting traction on the dura by endoscopic instruments or may be mistaken for patological fibrosis. These are only assumptions without any clear evidence. I suggest revising the study using at least one other method for investigating these ligaments (anatomical specimen, observations during surgical procedures, radiological methods..) to investigate these interesting assumptions. Also I suggested adding inclusion and exclusion criteria, indications for the procedures, and add basic demographic details about the patient cohort. Also add how many had previous interventions or surgeries in this region as this could definitely influence the presence of the lilgament.
I would also like to know at which vertebral levels was the epidural space investigated? Could the dorsomedian ligament be located only in certain levels and absent in others? Also, were the other epidural ligaments found? There are many of them and they have various prevalnces. Lastly, I recommend descirbing how you distinguish "pathological septa" from anatomical ligaments?
The quality of English is very poor. I am not able to understand some parts of the article with 100% certainty. Improvement of the language is not the job of an English proof reader, but of the authors, together with somebody proficient in English.